# Displacement Monitoring of a Bridge Based on BDS Measurement by CEEMDAN–Adaptive Threshold Wavelet Method

**DOI:** 10.3390/s23094268

**Published:** 2023-04-25

**Authors:** Chunlan Mo, Huanyu Yang, Guannan Xiang, Guanjun Wang, Wei Wang, Xinghang Liu, Zhi Zhou

**Affiliations:** 1School of Information and Communication Engineering, Hainan University, Haikou 570228, China; 2Faculty of Infrastructure Engineering, Dalian University of Technology, Dalian 116024, China; 3School of Computer Science and Technology, Hainan University, Haikou 570228, China; 4College of Civil Engineering and Architecture, Hainan University, Haikou 570228, China

**Keywords:** beidou navigation system (BDS), bridge monitoring, complete ensemble empirical mode decomposition with adaptive noise (CEEMDAN), adaptive threshold wavelet, data noise reduction

## Abstract

From the viewpoint of BDS bridge displacement monitoring, which is easily affected by background noise and the calculation of a fixed threshold value in the wavelet filtering algorithm, which is often related to the data length. In this paper, a data processing method of Complete Ensemble Empirical Mode Decomposition with Adaptive Noise (CEEMDAN), combined with adaptive threshold wavelet de-noising is proposed. The adaptive threshold wavelet filtering method composed of the mean and variance of wavelet coefficients of each layer is used to de-noise the BDS displacement monitoring data. CEEMDAN was used to decompose the displacement response data of the bridge to obtain the intrinsic mode function (IMF). Correlation coefficients were used to distinguish the noisy component from the effective component, and the adaptive threshold wavelet de-noising occurred on the noisy component. Finally, all IMF were restructured. The simulation experiment and the BDS displacement monitoring data of Nanmao Bridge were verified. The results demonstrated that the proposed method could effectively suppress random noise and multipath noise, and effectively obtain the real response of bridge displacement.

## 1. Introduction

Bridges are a vital component of the road infrastructure network and the development of the national economy. During operation, bridges may be affected by factors such as traffic loads, wind, and earthquakes, resulting in a certain amount of deformation of the bridge. In severe cases, it can lead to safety accidents. Therefore, it is necessary to monitor the health of bridges, especially the displacement generated by the bridge under load [1]. In addition to strain gauge [2], optical fiber sensor, and accelerometer [3], there are also precise level, robotic total station (RTS) [4,5,6,7,8,9], LiDAR DTMs [10,11], and global navigation satellite system (GNSS) instrument [12,13,14], etc. Strain gauge and optical fiber sensors have limitations in displacement measurement [1], and the calculation of accelerometer by double integration will lead to serious errors [15,16]. Although level and RTS have achieved good results in displacement monitoring, there are still some limitations, such as not working all day as well as the need to see between stations [12,17,18]. In contrast, as a geodetic survey method, GNSS technology has the advantages of providing three-dimensional coordinates, all-weather operation, and not requiring a line-of-sight between target points [13,14]. As early as 1997, GPS had been applied to the Humber Bridge, and the dynamic displacement of the bridge was successfully obtained [19], indicating that GPS could be applied to bridge monitoring. With the development of GNSS technology, its application in bridge displacement monitoring had also been greatly developed, such as with the application of GNSS technology in the Talkha highway steel bridge in Mansoura [20], the Yeongjong Grand Bridge in South Korea [21], and the Severn Suspension Bridge in the UK [22]. A large amount of literature has been published in recent years, but most of the research is on GPS [20,23]. Beidou Navigation Satellite System (BDS) is a satellite navigation system that has been formally networked in 2020, and has also been applied in the field of deformation monitoring [24,25]. In terms of bridge displacement monitoring, Xi et al., demonstrated through experiments that the performance of BDS-RTK in bridge monitoring is comparable to that of GPS-RTK, and BDS-RTK is subjected to less background noise [26]. This indicates the feasibility of BDS in bridge displacement monitoring.

When using GNSS for bridge displacement monitoring, due to environmental influences, GNSS signals may contain some noise, such as satellite multipath effects, random noise, etc. [27,28,29]. The impact of these noises will submerge the true displacement information of the bridge, leading to a decrease in the accuracy of bridge health monitoring. Therefore, it is necessary to find an appropriate method to filter these noises. Most traditional signal processing methods focus on signals with periodic stationarity, while the results of GNSS monitoring signals are nonlinear and non-stationary [30], which includes true displacement information of bridges, errors caused by multipath effects, and random noise [11]. Empirical mode decomposition (EMD) [31] and wavelet analysis [32] are two commonly used and effective methods to process nonlinear and non-stationary signals. Variational Mode Decomposition (VMD) has a good application effect in the signal processing of bridge displacement monitoring due to its unique advantages [33]. However, CiveraM et al., noted in their study that the VMD method is not applicable to non-stationary signals and has certain limitations, while the EMD is more suitable for non-stationary signals [34]. Mode aliasing will occur when a signal is decomposed by EMD [35]. Although EEMD can greatly eliminate modal mixing by adding white noise to EMD, the added noise cannot be completely removed [36]. CEEMDAN proposed by Torres et al. [37] is an improved CEEMD [38] with adaptive noise, which can achieve an accurate recombination of decomposed signals and effectively solve the problem of modal mixing. The CEEMDAN filtering method is used to remove components containing noise directly, but there would still be some effective information in the components, which would be lost if discarded directly. In order to avoid this situation, the EMD method, combined with other data processing methods, has been applied in GNSS signal de-noising. Therefore, Gao et al., used the Hilbert-Huang transform (HHT) and EEMD analysis methods to study the time-frequency characteristics of GNSS strain time series before the Yunnan earthquake, which could better analyze the change characteristics of seismic signals at different scales [39]. Wei et al., combined EEMD with independent component analysis (PSR-ICA) based on phase space reconstruction to analyze the vertical time series of GNSS reference stations and effectively separate independent atmospheric and soil moisture load signals [40]. Wen Chen et al., proposed a method combining the Chebyshev filter and CEEMDAN (CF-CEEMDAN) to de-noise GNSS monitoring signals of offshore platforms [41].

As a classical multi-scale analysis method, wavelet analysis plays an important role in GNSS signal noise reduction [42,43,44]. In addition to using wavelet analysis alone to process GNSS signals, the GNSS signals are also filtered jointly with the empirical mode decomposition method to solve the detail loss caused by empirical mode decomposition method. For example, RuiRao et al., proposed to use EMD combined with wavelet analysis to de-noise bridge GNSS monitoring and correctly extract bridge frequency [45]. However, this algorithm is affected by the problem of mode mixing in EMD. Niu et al., proposed that EEMD combined the wavelet packet method for dynamic analysis of suspension bridges, and proved that the proposed EEMDWP method was proved to be better than the single EEMD or WP method [46]. Guo et al., proposed a filtering method combining EEMD and wavelet analysis to separate multipath effects in GNSS data [47]. Xiong et al., used CEEMDAN combined with wavelet transform [48] or wavelet packet [30] to conduct noise reduction processing on GNSS-RTK monitoring data of bridges and high-rise buildings. Although the noise reduction method of empirical mode decomposition combined with wavelet analysis solved the problem of detail information loss to a certain extent, the filtering effect of wavelet analysis method depended on the selection of wavelet basis function, decomposition layers, and threshold. The selection of threshold is usually a fixed threshold or a threshold parameter based on a large amount of experience, which had certain subjectivity. Different threshold selection had different noise reduction effects.

Based on the above analysis, in order to correctly identify the bridge displacement, change in the BDS displacement monitoring signal containing noise, and to reduce the influence of subjectivity of threshold selection in wavelet de-noising, a method combining CEEMDAN and adaptive threshold wavelet (CEEMDAN-AWT) is proposed in this paper to reduce the noise of bridge BDS monitoring signal. The Section 2 introduces the method of data collection in this paper. In Section 3, the principle of this method is introduced and the performance of this method is evaluated by using analog signal. In Section 4, firstly, the stability of the BDS receiver is tested, and the noise reduction effect of the proposed method on the measured data is evaluated. Finally, the monitoring signals of bridge engineering are applied and the monitoring results of bridge displacement are analyzed.

## 2. Data Collection Method

The equipment used for data collection in this paper is the split GNSS receiver M900SE provided by Sinan Navigation Company, as shown in Figure 1. The antenna is an AT60 antenna, which can simultaneously observe the entire galaxy (GPS, GLONASS, Galileo, BDS) satellite. The receiver is equipped with a 4G transmission module, which can upload the collected data to the cloud server in real time through the wireless network. Simultaneously, it is also equipped with an RS232 serial port, which can transmit the data through the wired way, which allows the data collection to be more comprehensive and convenient. The data collection flow chart is shown in Figure 2.

The steps of data collection are as follows:A BDS receiver is used to receive signals from BDS satellites.

The observation method adopted in this paper is relative positioning technique [14]. Relative positioning refers to the joint observation of the satellite by the reference station and the monitoring station. The coordinates of the reference station are usually known by certainty, and then the coordinates of the monitoring station are determined by calculating the baseline vector between the reference station and the monitoring station. This occurs as shown in Figure 3.

In Figure 3, S1–S3 represent the satellites. T1 and T2 are two receiving devices, one of which serves as the reference station and the other as the monitoring station.
2.After processing the received satellite signal, the receiving device transmits it to the cloud server through the 4G network or to the upper computer through an RS232 serial port.3.The collected satellite data is relative computed with the commercial software compass solution to obtain the three-dimensional coordinates of the monitoring station, and then the displacement time series is obtained through the Equation (1) [21].


(1)$$\left[\begin{array}{c} ∆x_{i}\\ ∆y_{i}\\ ∆z_{i} \end{array}\right]=\left[\begin{array}{c} x_{i}\\ y_{i}\\ z_{i} \end{array}\right]-\frac{1}{n}\sum_{i=0}^{i=n}\left[\begin{array}{c} x_{i}\\ y_{i}\\ z_{i} \end{array}\right]$$
where $\left(∆x_{i},\;∆y_{i},\;∆z_{i}\right)\;$ is the displacement of each recorded moment (epoch), *n* is the total number of epochs, *i* = 1, 2, 3.

## 3. Principle of CEEMDAN-Adaptive Threshold Wavelet Algorithm

This section introduces the basic theory and workflow of CEEMDAN-adaptive threshold wavelet (CEEMDAN-AWT) method and tests the method with analog signals. Firstly, the signal is decomposed by the CEEMDAN method, and the noisy component and effective component are distinguished by the correlation coefficients between each component and the signal. Then, adaptive threshold wavelet de-noising is carried out on the noisy component to retain more details of the signal. Finally, all components are reconstructed.

### 3.1. CEEMDAN Algorithm

Adaptive noise complete set Empirical Mode decomposition (CEEMDAN) is an improved algorithm for the EMD [31] algorithm and EEMD [36] algorithm with mode aliasing and residual white noise in components. The CEEMDAN decomposition steps of bridge monitoring signal *y(t)* [37] are as follows:
A new signal $y^{\prime }\left(t\right)=y\left(t\right)+{\left(-1\right)}^{q}\varepsilon v^{j}\left(t\right)$ is obtained by adding Gaussian white noise to *y*(*t*).
where *q =* 1, 2, ε is the standard deviation of white noise, $v^{j}\left(t\right)$ is the Gaussian white noise signal, *N* is the number of times white noise is added, *j =* 1, 2.... After EMD decomposition of the new signal, the first order characteristic mode component *C*_1_ of EMD decomposition is obtained.
(2)$$E\left(y\left(t\right)+{\left(-1\right)}^{q}\varepsilon v^{j}\left(t\right)\right)=C_{1}^{j}\left(t\right)+r^{j}$$
where the function *E*($y\left(t\right)+{\left(-1\right)}^{q}\varepsilon v^{j}\left(t\right)$) represents the EMD decomposition process, $C_{1}^{j}\left(t\right)$ is the first order eigenmode component of EMD decomposition, and $r^{j}$ is the residual of EMD decomposition.
2.The first modal component IMF_1_ of CEEMDAN decomposition is obtained by an overall averaging of the *N* modal components generated:
(3)$$IMF_{1}=\bar{C_{1}\left(t\right)}=\frac{1}{N}\sum_{j=1}^{N}C_{1}^{j}\left(t\right)$$
where *N* is the number of first-order modal components decomposed by EMD.
3.Remove the first modal component to obtain the first residual of the signal:
(4)$$r_{1}\left(t\right)=y\left(t\right)-IMF_{1}\;$$
where $y\left(t\right)$ is the original signal added with white noise and ${\mathrm{IMF}}_{1}$ is the first-order modal component of CEEMDAN.
4.A new signal $r_{1}^{’}\left(t\right)$ is obtained by adding white Gaussian noise to $r_{1}\left(t\right)$, and then the signal is decomposed by EMD to obtain the first mode component *D*_1_ of $r_{1}^{’}\left(t\right)$, and the second eigenmode component ${\mathrm{IMF}}_{2}$ is obtained by averaging:

(5)$$IMF_{2}=\bar{C_{2}\left(t\right)}=\frac{1}{N}\sum_{j=1}^{N}D_{1}^{j}\left(t\right)\;$$
where *N* is the number of first-order modal components decomposed by EMD, and $D_{1}^{j}\left(t\right)$ is the first-order modal components decomposed by signal $r_{1}^{’}\left(t\right)$.
5.After removing the second modal component, the second residual $r_{2}\left(t\right)$ of signal y(t) is obtained:

(6)$$r_{2}\left(t\right)=r_{1}\left(t\right)-{\mathrm{IMF}}_{2}\;$$
where $r_{1}\left(t\right)$ is the first residual of the signal *y*(*t*) and ${\mathrm{IMF}}_{2}$ is the second CEEMDAN modal component of the signal *y*(*t*).
6.Repeat the above steps until the obtained residual signal is a monotone function, then the decomposition is complete. At this point, the signal *y*(*t*) is decomposed into *K* modal components and a residual component:

(7)$$y\left(t\right)=\sum_{k=1}^{K}IMF_{k}+r_{t}\left(t\right)\;$$
where *K* is the number of modal components, ${\mathrm{IMF}}_{k}$ is the *K*th modal component, and $r_{t}\left(t\right)$ is the residual component of signal *y*(*t*).

It can be seen from Equation (7) that bridge displacement monitoring signal *y*(*t*) can be decomposed into *K* modal components and one residual term. The CEEMDAN method directly removes the noise component to realize signal de-noising.

### 3.2. Adaptive Threshold Wavelet Algorithm

The CEEMDAN method directly removes the noisy component, but will lose some details of the signal. In contrast, the wavelet threshold de-noising method is de-noising the signal through the limit of the threshold. Therefore, wavelet threshold de-noising can retain more effective information in the noisy component. The specific methods of wavelet threshold de-noising for noisy components are as follows:Select the appropriate wavelet basis function and the appropriate number of decomposition layers.Quantify the high-frequency decomposition layer with a threshold value.All wavelet coefficients are reconstructed to obtain de-noised signals.

In the above steps, the selection of threshold is the key link in wavelet threshold de-noising. Common thresholds include an unbiased risk estimation threshold, fixed threshold, heuristic threshold, and minimax threshold [42], among which the fixed threshold is widely used in GNSS signal de-noising, and can be presented as follows [45]:(8)$$\lambda =\sigma \sqrt{2lnN}\;\sigma =\frac{median\left(\left|cd1\right|\right)}{0.6745}$$
where *σ* is noise variance, *N* is signal length, and *cd*1 is the detail coefficient of the first layer decomposition [45].

According to Equation (8), the fixed threshold is not only related to noise variance, but also to signal length. If the signal is too long or too short, the de-noising effect will be weakened to some extent. In view of the above situation, this paper proposes an adaptive threshold calculation method, and can be presented as follows:(9)$$\lambda =\mu_{j}+max\left(cd_{j}\right)*\delta_{j}\;j=1,2,\ldots ,k\;$$
where *μ* is the mean value, *δ* is the variance of the wavelet coefficients of this layer, respectively, and *j* is the number of decomposition layers. The threshold calculation method is to adaptively select the high frequency coefficients of each layer to avoid the influence of data length on the threshold and all wavelet coefficients use the same threshold filtering.

### 3.3. CEEMDAN-Adaptive Threshold Wavelet Algorithm Process

The noise reduction process of CEEMDAN-AWT method is shown in Figure 4. The specific process is as follows:The original signal was decomposed into each order modal component (IMF component) and a residual component (res component) by CEEMDAN.Calculate the correlation coefficients *r* between IMF components of each order and the original signal [30], and it can be presented as follows:
(10)$$r=\frac{\sum_{i}^{N}\left(y\left[i\right]-\bar{y}\right)\left(IMF\left[i\right]-\bar{IFM}\right)}{\sqrt{\sum_{i}^{N}{\left(y\left[i\right]-\bar{y}\right)}^{2}{\left(IMF\left[i\right]-\bar{IFM}\right)}^{2}}}\;$$
where *y* is the original data, $\bar{y}$ is the average value, IMF is the modal components of each order, and $\bar{IFM}$ is the average value. The IMF component with the first local minimum of the correlation coefficient is selected as the boundary, the IMF component from the first IMF component to this IMF component is the noise component, and the remaining components are the effective components.
3.The noisy components are de-noised by the wavelet threshold and quantized by the soft threshold function [45]. The soft threshold function is as follows:
(11)$$w_{\lambda }=\left\{\begin{array}{c} sgn\left(w\right)*\left(\left|w\right|-\lambda \right)\;\left|w\right|\geq \lambda \;\\ 0\;\left|w\right|<\lambda \end{array}\right.\;$$
where *w* is each wavelet coefficient, λ is the critical threshold, and $sgn\left(x\right)$ is a symbolic function.

4.The filtered IMF components was reconstructed.

In order to evaluate the noise reduction performance of the noise reduction method, signal-to-noise ratio (SNR) and root mean square error (RMSE) are introduced [30]. As follows:(12)$$SNR\;=10log10\frac{\sum_{i}^{N}y{\left(i\right)}^{2}}{\sum_{i}^{N}{\left[y\left(i\right)-y^{\prime }\left(i\right)\right]}^{2}}\;$$
(13)$$RMSE\;=\sqrt{\frac{1}{N}\sum_{i}^{N}{\left[y\left(i\right)-y^{\prime }\left(i\right)\right]}^{2}}\;$$
where $y\left(t\right)$ is the original data, $y^{\prime }\left(t\right)$ is the data after noise removal, and *N* is the data length. The larger the SNR or the smaller the RMSE denotes the better the noise reduction effect.

### 3.4. Performance Evaluation of CEEMDAN-Adaptive Threshold Wavelet Algorithm

To evaluate the performance of CEEMDAN-AWT method, $y\left(t\right)=5sin\left(2\pi *0.7t\right)+7sin\left(2\pi *0.5t\right)+n\left(t\right)$ was used as the analog signal in this paper. The signal consists of sinusoidal signals with frequency of 0.7 Hz and 0.5 Hz and random noise *n(t)*. The sampling frequency of the analog signal is 100 Hz. Figure 5 shows the images of signal without noise and signal *y*(*t*) with 5 dB noise added.

By comparing Figure 5a,b, it can be seen that after adding noise, the signal appears burr and the signal smoothness is weakened, indicating that noise will affect the correct recognition and use of the signal, therefore, it needs to be processed by filtering. After CEEMDAN of *y*(*t*), 11 IMF components and 1 residual component were shown in Figure 6. Equation (10) was used to calculate the correlation coefficients between each IMF component and *y*(*t*), as shown in Table 1.

It can be seen from Table 1 that the first locally minimum IMF component of the correlation coefficient was IMF4, as such it was determined that IMF1~IMF4 were noise components, while IMF5~IMF11 were effective components. The comparison between the reconstructed signal and *y*(*t*) after processing by using the CEEMDAN-AWT method proposed in this paper is shown in Figure 7.

It can be seen from Figure 7 that after filtering, the signal was smoother and less sharp, which was more similar to *x*(*t*) and retained the details of the signal. In order to evaluate the noise reduction effect of the proposed method, it was compared with the CEEMDAN method and CEEMDAN-fixed threshold wavelet (CEEMDAN-FWT) method, respectively. The evaluation indexes calculated according to Equations (12) and (13) were shown in Table 2. In Table 2, noise signals at different signal-to-noise ratio levels (5 dB, 10 dB, 15 dB, 20 dB, and 25 dB) were also compared and analyzed.

It can be seen from Table 2 that the SNR and RMSE of the signal processed by CEEMDAN-AWT method were the maximum and the minimum. This indicates that the performance of the CEEMDAN-AWT method was better than the other two methods and was more suitable for the suppression of random noise in BDS monitoring signals.

## 4. Experimental Results and Discussion

### 4.1. Background Noise Analysis

In order to evaluate the background noise of the receiver, two M900SE GNSS receivers of Sinan Navigation were used for static test in an open square. The experimental test Figure 8 was as follows. One receiver was used as the reference station and the other as the monitoring station. The two devices were used for synchronous data acquisition with a sampling frequency of 1 Hz and a total test time of 3 h and 33 min. The length of the baseline between the two stations was 4.06 m, which was classified as a short baseline (<5 km) [49]. Both sensors were stationary. Theoretically, the displacement of the monitoring station should have been zero. Therefore, the non-zero results in the test were visually generated by the background noise. The solution software was used to solve the collected BDS satellite signals, and the displacement time series was calculated according to Equation (1), as shown in Figure 9. The displacement time series included horizontal direction (north-south direction and east-west direction; N-S direction and E-W direction) and vertical direction (U direction). The mathematical statistical characteristics of the displacement time series in the three directions are shown in Table 3.

It can be seen from Figure 9 and Table 3 that the displacement data of the three directions were mainly concentrated near zero, in which the maximum value of the N-S direction was 16.926 mm and the minimum value was −37.074 mm; the maximum value of the E-W direction was 44.883 mm and the minimum value was −32.117 mm. The maximum value of the U direction was 19.080 mm and the minimum value was −15.920 mm. It can be seen from the Figure 9 that the maximum value occupied a small proportion column in the whole signal and did not appear continuously, and thus it was determined that there may have been gross error in this time series. In order to judge and remove gross errors, the commonly used 3σ criterion method was introduced, which was as follows [50]:
Calculate the average $\bar{X}$ of displacement time series, as shown in Equation (14).

(14)$$\bar{X}=\frac{1}{n}\sum_{i=1}^{n}X_{i}$$
where $\bar{X}$ is the mean value of the displacement time series, *n* is the sequence length, and $X_{i}$ is the displacement at every moment.
2.The residual error $v_{i}$ of the sequence is calculated as shown in Equation (15).


(15)$$v_{i}=X_{i}-\bar{X}$$
where $X_{i}$ is the displacement at every moment, and $\bar{X}$ is the mean value of the displacement time series.
3.Calculate the root mean square deviation, σ, of the sequence according to Bessel method, as shown in Equation (16).


(16)$$\sigma =\sqrt{\sum v_{i}^{2}/\left(n-1\right)}$$
where $v_{i}$ is the residual of the monitoring data sequence and *n* is the length of the sequence.
4.Judge according to the above results. If $\left|X_{i}-\bar{X}\right|>3\sigma$, $X_{i}$ will be judged as gross error and removed, and the average value will be used for interpolation; otherwise, $X_{i}$ will be judged as normal value and retained.


After removing coarse error, the horizontal displacement ranged from −12.074~11.926 mm in the north-south direction and from −20.117~19.883 mm in the east-west direction. The vertical displacement was −8.920~9.080 mm. The accuracy of the BDS sensor used in this study was ±10 mm in the horizontal direction and ±15 mm in the vertical direction. The displacement generated by environmental noise exceeded the measurement range in the horizontal direction, while in the vertical direction, although it did not exceed the measurement accuracy, it could be seen from the figure that there were some fluctuations and burrs in the displacement time series, which still indicated that the measurement by BDS sensor would be affected by noise. The displacement time series was processed by CEEMDAN-AWT filtering. Figure 10 demonstrated the IMF components of displacement time series in three directions. The correlation coefficients, *r*, of each IMF component and each displacement time series were calculated, respectively, as shown in Table 4.

According to Table 4, the correlation coefficient 0.125 of IMF5 in the north-south direction was the first local minimum, while the first local minimum of the correlation coefficient in the east-west direction and the vertical direction was IMF4, which was 0.178 and 0.204, respectively. Therefore, in the N-S direction IMF1-IMF5 were noise components and IMF6-IMF11 were effective components. In the E-W and U directions IMF1-IMF4 were noise components and IMF5-IMF11 were effective components. The comparison between the filtered displacement time series and the original displacement time series was shown in Figure 11. As can be seen from Figure 11, after noise reduction the time series became smoother, the sharp points were suppressed, and the data accuracy was also improved. Under static observation after noise reduction, the displacement range in the N-S direction was −9.915~11.180 mm, the displacement range in the E-W direction was −15.856~15.833 mm, and the displacement range in the U direction was −7.409~7.780 mm. It can be seen that in addition to the east-west direction, the measurement accuracy of the instrument was satisfied in the north-south direction and the vertical direction. Under the influence of the multipath effect, the accuracy of the U direction was still less than 8mm, which could meet the needs of bridge monitoring.

In order to evaluate the noise reduction effect of the proposed method on static test data, it was compared with the CEEMDAN method and the CEEMDAN-FWT method, then the evaluation indexes of the three methods were calculated using Equations (12) and (13), as shown in Table 5.

It can be seen from Table 5 that SNR of the CEEMDAN-AWT method for displacement time series de-noising in three directions in a static test was the largest among the three methods, while RMSE was the smallest among the three methods. This meant that the performance of the CEEMDAN-AWT method in actual measurement data was better than the other two algorithms.

### 4.2. De-Noising of Bridge BDS Displacement Measurement Signal

A field survey was carried out in Nanmaoqiao, Baoting Autonomous County, Hainan Province. The length of Nanmao bridge was 260.68 m. The main bridge was 127.0 m long and 15 m wide, which was a highway bridge connecting Baoting County and Nanmao Farm. The bridge was the only way for people to travel and purchase farm, and the traffic flow was relatively large. In this test, a monitoring station was set up at the mid-span position of the bridge, and a reference station was set up on an unopened road 91 m away from the bridge. The coordinates of the reference station were obtained by conventional static positioning method. The instruments used in the test included two split receivers of Sinan M900SE, equipped with power supply equipment and laptop computers. Figure 12 demonstrates the layout of monitoring stations and reference stations in this test. The experiment occurred at 13:00 on 8 December 2022, with a total of five hours of observation. The sampling frequency of the equipment was 1 Hz, and the cut-off angle of the satellite was 15 degrees. Since the bridge was mainly affected by the traffic volume, this paper only analyzed the time series in the vertical (U) direction. Figure 13 showed the displacement time series diagram in the vertical direction of this test. Table 6 showed the basic statistical characteristics of bridge monitoring data.

After calculation, the maximum value and minimum value of the data obtained in this experiment was 28.969 mm and −46.031 mm. However, as shown in Figure 13, these data occupy relatively few specific columns throughout the entire time series and were not clustered. Therefore, it was determined that there was a coarse error in the displacement time series, and the 3σ method performed coarse error processing on the original data. The baseline length between the two stations in this test was 91 m, which was considered a short baseline. The satellite clock deviation and receiver clock deviation could be eliminated through relative positioning, and the atmospheric propagation delay could be ignored [50]. The main source of error in the displacement time series was the random noise received by the device during reception. The displacement time series after gross error removal was decomposed by CEEMDAN, and the IMF components were shown in Figure 14. The correlation coefficient, *r*, of each IMF component and displacement time series was calculated according to Equation (10), as shown in Table 7.

As can be seen from Table 7, the first local minimum of correlation coefficient was IMF5, and its correlation coefficient with displacement time series was 0.145. Therefore, IMF~IMF5 was judged as the noise component, and IMF6~IMF12 as the effective component. The adaptive threshold wavelet de-noising was carried out for IMF1-IMF5 and reconstructed with the effective component. The comparison between the filtered displacement time series and the original displacement time series was shown in Figure 15. In order to further evaluate the noise reduction effect of the proposed method on displacement time series, the CEEMDAN method and the CEEMDAN-FWT method were carried out on displacement time series at the same time, and their evaluation indexes were calculated and shown in Table 8.

As shown in Table 8, after noise reduction by the CEEMDAN-AWT method, SNR of displacement time series was 7.186 dB, which was the largest among the three methods, and RMSE was 2.884 mm, which was the smallest among the three methods. This meant that the noise reduction effect of the proposed method was better than that of the other two methods, which was consistent with the simulation results.

It can be seen from Figure 15 that the displacement time series became smoother and less sharp after de-noising. The correlation coefficient between the calculated filtered displacement time series and the original displacement time series was 0.899. The higher correlation meant that more details of the displacement response of the bridge were preserved, which can better reflect the deformation information of the bridge. According to Figure 15, the maximum value of displacement time series after noise reduction was 13.595 mm and the minimum value was −16.179 mm. The period of maximum displacement was about 12.500 s to 15.000 s in the time series, which was from 16:00 to 18:00 of the day. During this period, it was the peak time of commuting and the traffic volume was relatively large.

## 5. Conclusions

In this paper, a CEEMDAN-AWT method is proposed to solve the problem that the monitoring data of bridge displacement monitoring by BDS technology will be submerged due to the influence of background noise. The following conclusions are obtained through the test:

Through the SNR and RMSE analysis of three proposed methods of CEEMDAN, CEEMDAN-FWT method, and CEEMDAN-AWT method for a series of analog signals with different SNR levels, it is found that the CEEMDAN-AWT method has a better noise reduction effect than other two methods. It can be used to improve the precision of BDS displacement monitoring.

The stability test data of BDS receiver is processed by the CEEMDAN-AWT method. After noise reduction, the SNR of the north-south signal is 7.472 dB and RMSE of 1.751 mm, and the SNR of east-west signal is 6.325 dB and RMSE of 3.265 mm. The SNR and RMSE of vertical signal were 6.393 dB and 1.556 mm, respectively. The random noise of the three direction monitoring signals is suppressed. In the horizontal direction, the measuring range of the north-south direction is −9.915~11.180 mm, and the measuring range of the east-west direction is −15.856~15.833 mm, which is still higher than the measuring accuracy of the instrument after filtering, so it is necessary to seek a better method for processing. The measurement range in the vertical direction is −7.409~7.780 mm, which meets the monitoring requirements in the vertical direction of the bridge.

After using the proposed CEEMDAN-AWT method to reduce bridge data, the SNR is 7.186 dB and RMSE is 2.884 mm, and the noise reduction effect is better than the other two methods. The correlation coefficient between the filtered monitoring data and the original monitoring data is 0.899, which effectively preserves the detailed information of the displacement monitoring response signal. During the monitoring period, the maximum displacement change of −16.179 mm occurred under the traffic load. This method provides an excellent noise reduction method for bridge deformation monitoring, but it is a pity that the dynamic response of the bridge cannot be analyzed deeply due to the limitation of sampling frequency of experimental instruments.

## Figures and Tables

**Figure 1 sensors-23-04268-f001:**
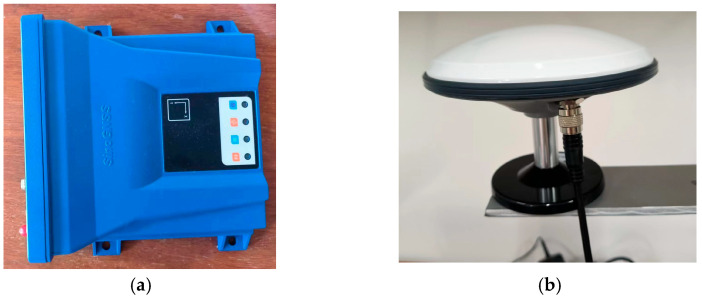
Main equipment of data collection: (**a**) M900SE receiver; (**b**) AT60 antenna.

**Figure 2 sensors-23-04268-f002:**
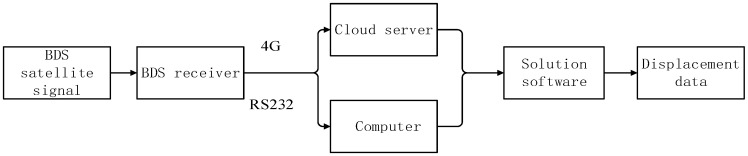
Data collection flow chart.

**Figure 3 sensors-23-04268-f003:**
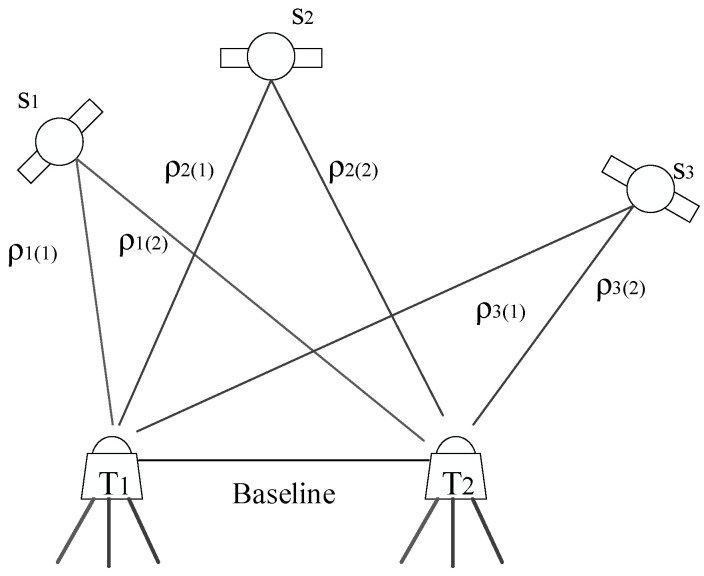
Relative positioning work diagram.

**Figure 4 sensors-23-04268-f004:**
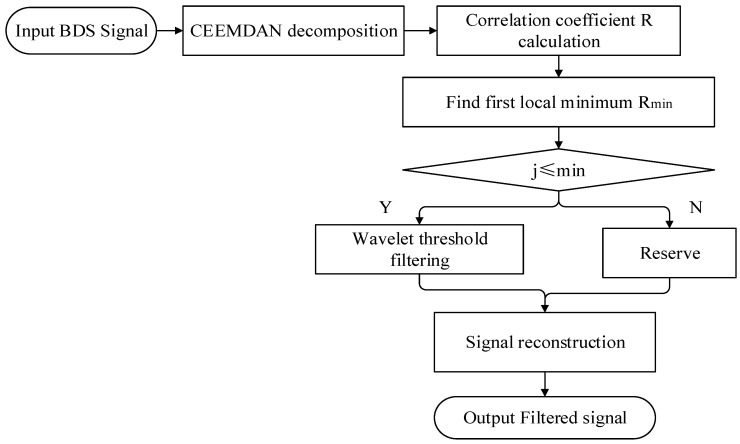
Flow chart of noise reduction.

**Figure 5 sensors-23-04268-f005:**
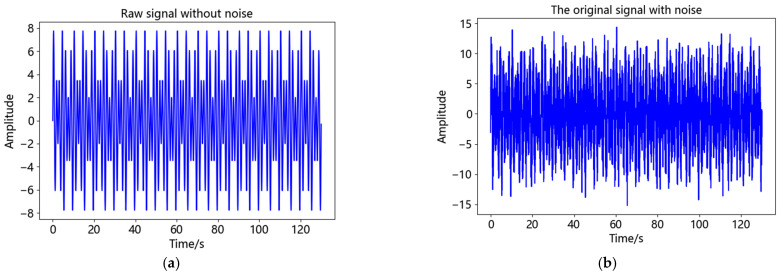
Amplitudes of *x*(*t*) and *y*(*t*): (**a**) Amplitudes of *x*(*t*); (**b**) Amplitudes of *y*(*t*).

**Figure 6 sensors-23-04268-f006:**
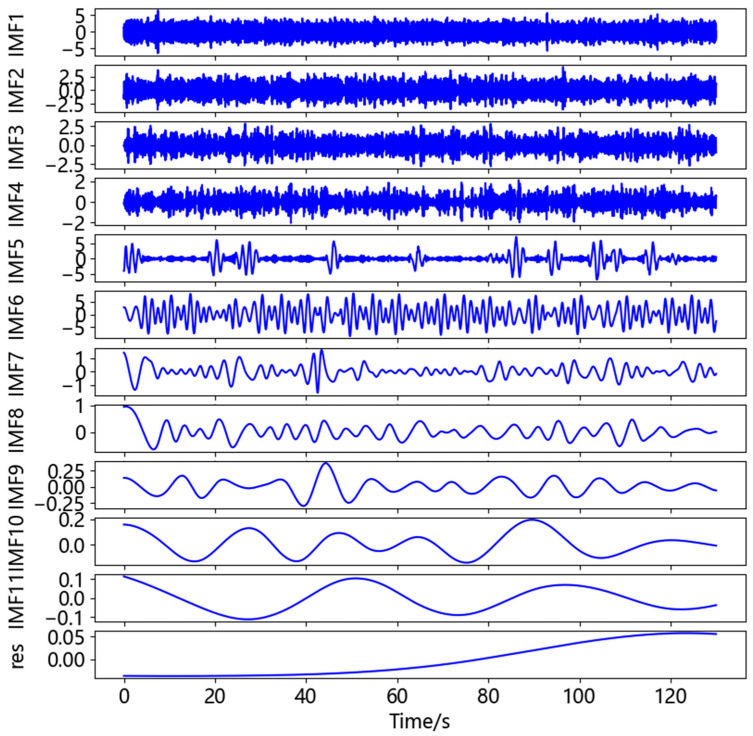
CEEMDAN decomposition of *y*(*t*).

**Figure 7 sensors-23-04268-f007:**
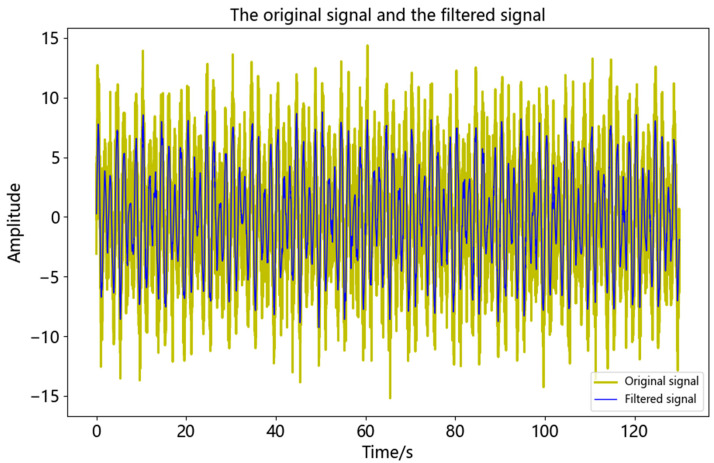
Comparison between the analog signal and the original signal after noise reduction.

**Figure 8 sensors-23-04268-f008:**
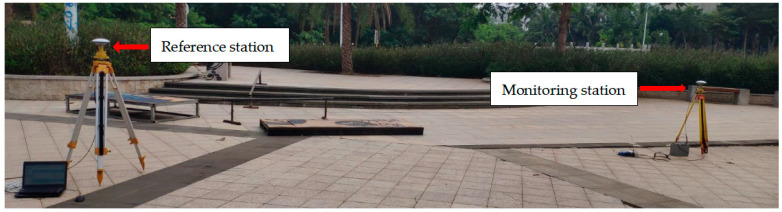
Background noise test system diagram.

**Figure 9 sensors-23-04268-f009:**
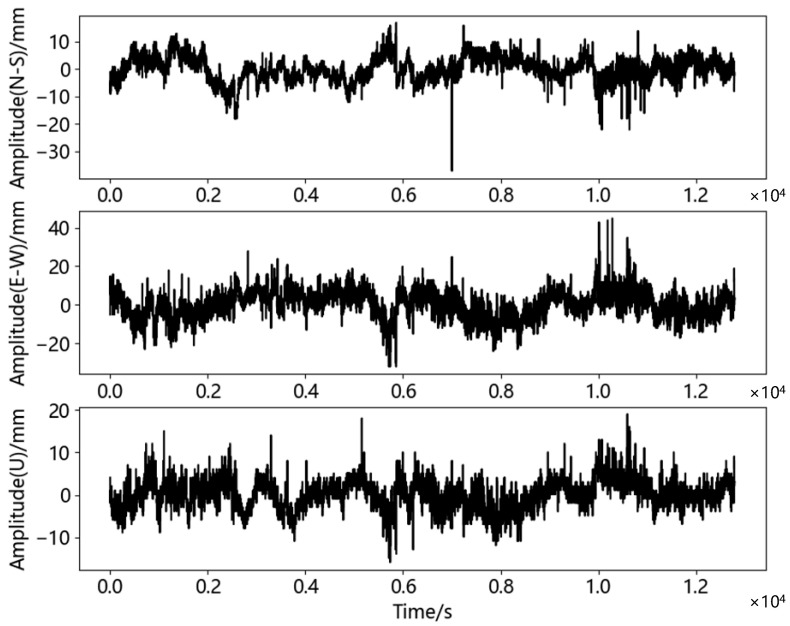
Three direction original time series of the BDS static test.

**Figure 10 sensors-23-04268-f010:**
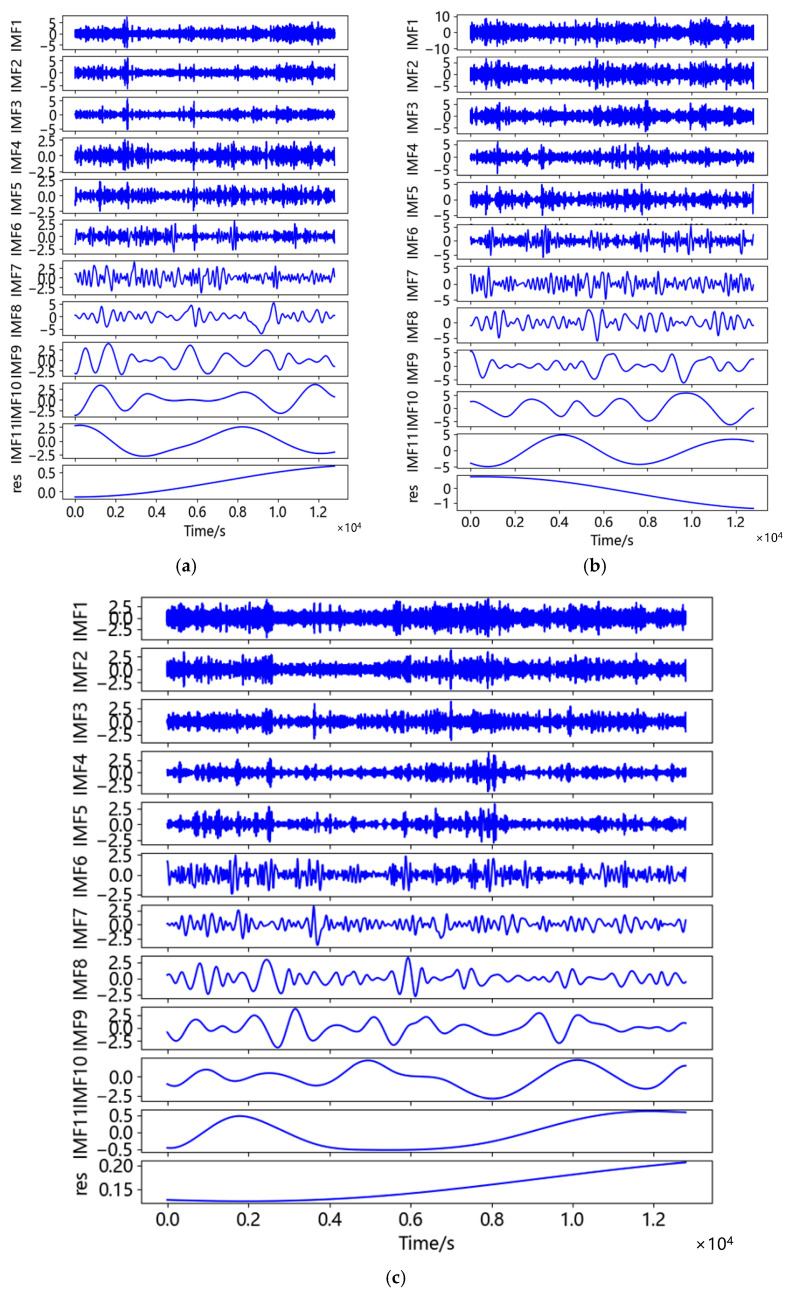
CEEMDAN decomposition of data in three directions: (**a**) CEEMDAN decomposition of data in N-S direction; (**b**) CEEMDAN decomposition of data in E-W direction; (**c**) CEEMDAN decomposition of data in U direction.

**Figure 11 sensors-23-04268-f011:**
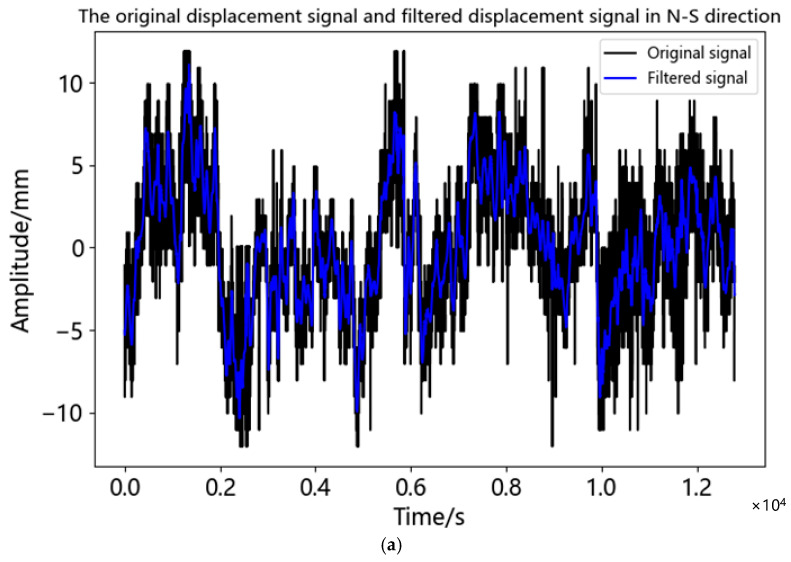
Comparison of original and de-noised signals in three directions of static test experiment: (**a**) Comparison of original and de-noised signals in N-S direction; (**b**) Comparison of original and de-noised signals in E-W direction; (**c**) Comparison of original and de-noised signals in U direction.

**Figure 12 sensors-23-04268-f012:**
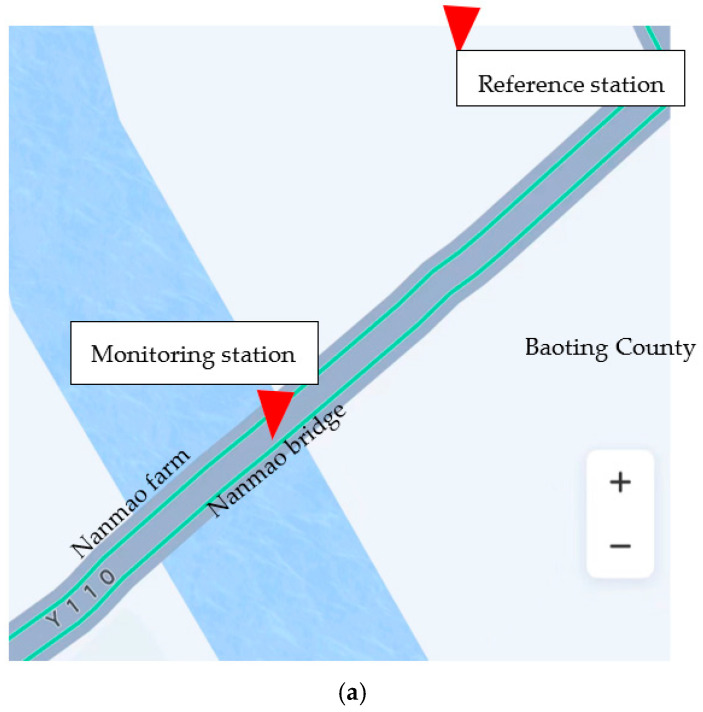
Layout diagram of bridge testing sensors: (**a**) Sensor’s layout diagram (background image from Baidu Maps); (**b**) Reference station; (**c**) Monitoring station.

**Figure 13 sensors-23-04268-f013:**
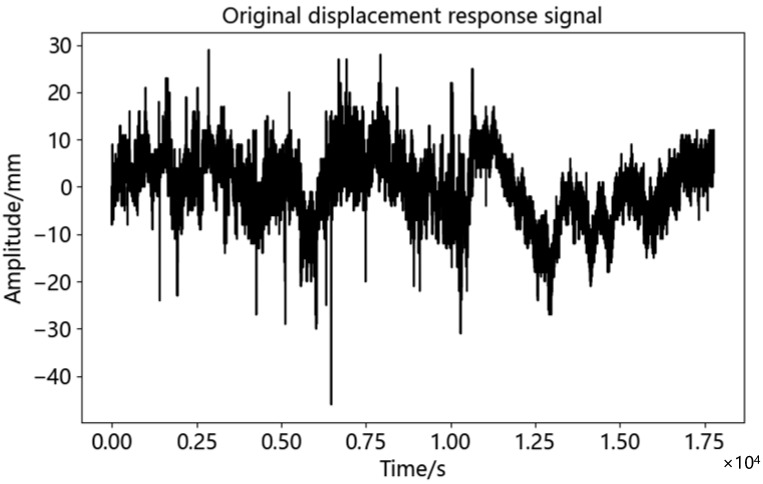
Original time series of elevation displacement.

**Figure 14 sensors-23-04268-f014:**
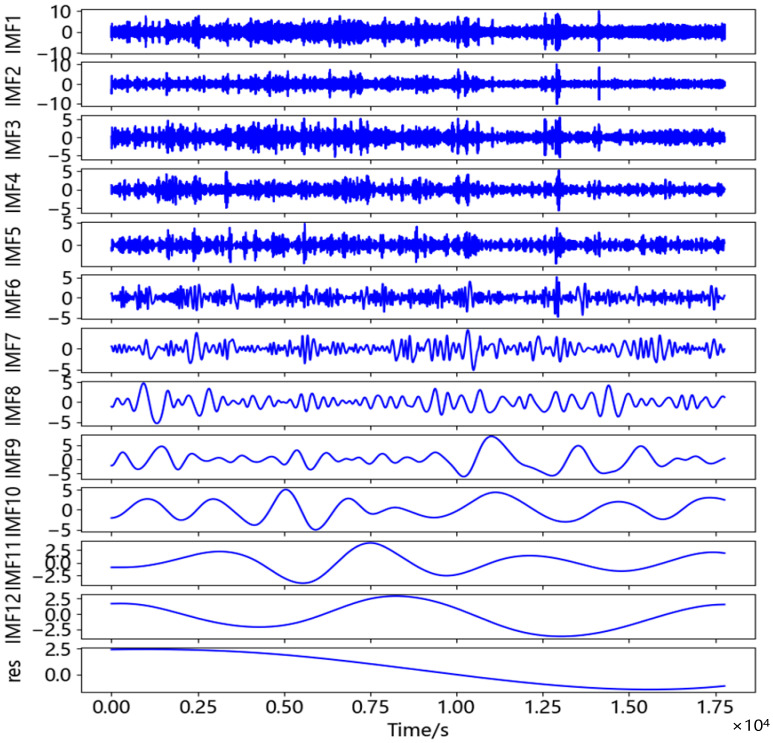
CEEMDAN decomposition diagram in U direction of Nanmao Bridge dynamic test.

**Figure 15 sensors-23-04268-f015:**
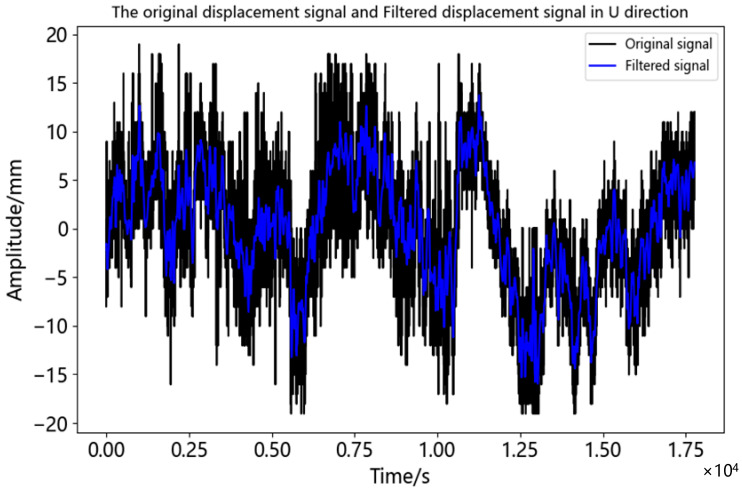
Comparison of the original signal and de-noised signal of bridge displacement monitoring.

**Table 1 sensors-23-04268-t001:** Correlation coefficients between IMF components and *y*(*t*).

IMF No.	IMF1	IMF2	IMF3	IMF4	IMF5	IMF6	IMF7	IMF8	IMF9	IMF10	IMF11
Correlation coefficients	0.357	0.211	0.168	0.109	0.321	0.838	0.229	0.025	0.013	0.012	−0.014

**Table 2 sensors-23-04268-t002:** Evaluation indicators of analog signal noise reduction.

Different Noise Levels	5 dB		10 dB		15 dB		20 dB		25 dB	
	SNR/dB	RMSE/mm	SNR/dB	RMSE/mm	SNR/dB	RMSE/mm	SNR/dB	RMSE/mm	SNR/dB	RMSE/mm
CEEMDAN	6.356	2.271	10.662	1.265	15.368	0.715	20.609	0.386	25.539	0.218
CEEMDAN-FWT	6.360	2.269	10.698	1.260	15.527	0.702	20.891	0.374	26.030	0.206
CEEMDAN-AWT	6.377	2.265	10.786	1.247	15.857	0.676	22.410	0.314	30.188	0.127

**Table 3 sensors-23-04268-t003:** Basic information of the original time series.

Signal	Mean/mm	Std/mm	Max/mm	Min/mm
N-S direction	5.4180 × 10^−8^	4.3992	16.926	−37.074
E-W direction	−9.0183 × 10^−7^	7.0044	44.883	−32.117
U direction	−5.1194 × 10^−8^	5.4179	19.080	−15.920

**Table 4 sensors-23-04268-t004:** Correlation coefficients between IMF and original signals in three directions.

Signal	Correlation Coefficients										
	IMF1	IMF2	IMF3	IMF4	IMF5	IMF6	IMF7	IMF8	IMF9	IMF10	IMF11
N-S direction	0.256	0.182	0.162	0.151	0.125	0.239	0.334	0.415	0.520	0.404	0.371
E-W direction	0.324	0.220	0.206	0.178	0.193	0.256	0.264	0.252	0.311	0.432	0.523
U direction	0.308	0.229	0.209	0.204	0.208	0.215	0.219	0.347	0.525	0.271	0.342

**Table 5 sensors-23-04268-t005:** Noise reduction performance of different methods.

Signal	N-S Direction		E-W Direction		U Direction	
	SNR/dB	RMSE/mm	SNR/dB	RMSE/mm	SNR/dB	RMSE/mm
CEEMDAN	7.424	1.761	6.317	3.268	6.327	1.568
CEEMDAN-FWT	7.453	1.755	6.321	3.266	6.349	1.564
CEEMDAN-AWT	7.472	1.751	6.325	3.265	6.393	1.556

**Table 6 sensors-23-04268-t006:** Basic statistical characteristics of bridge monitoring data.

Signal	Mean/mm	Std/mm	Max/mm	Min/mm
U dir	−1.869 × 10^−12^	6.872	28.969	−46.031

**Table 7 sensors-23-04268-t007:** Correlation coefficient between IMF and original signal.

IMF No.	IMF1	IMF2	IMF3	IMF4	IMF5	IMF6
Correlation coefficients	0.279	0.190	0.171	0.152	0.145	0.155
IMF No.	IMF7	IMF8	IMF9	IMF10	IMF11	IMF12
Correlation coefficients	0.150	0.247	0.272	0.535	0.545	0.481

**Table 8 sensors-23-04268-t008:** Noise reduction performance of different methods of bridge monitoring signal.

Indicator	Method		
	CEEMDAN	CEEMDAN-FWT	CEEMDAN-AWT
SNR/dB	7.175	7.177	7.186
RMSE/mm	2.888	2.887	2.884

## Data Availability

The authors confirm that the data supporting the findings of this study are available within the article.

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
