# Peer review of "Displacement Monitoring of a Bridge Based on BDS Measurement by CEEMDAN–Adaptive Threshold Wavelet Method"

_sensors, 2023, doi:10.3390/s23094268_

Round 1
Reviewer 1 Report
The paper
“Displacement monitoring of a bridge based on BDS measurement by CEEMDAN-adaptive threshold wavelet method”,
By Chunlan Moet al.,
Presents an approach for bridge displacement monitoring, based on a CEEMDAN-adaptive threshold wavelet method. The methodology is tested and validated on synthetic and experimental data
The content of the manuscript is in line with the aim of the journal and can be considered for potential publication.
Nevertheless, some content and presentation issues need to be addressed before acceptance. Specifically:
1. in Fig 3 (numerically simulated signal), IMFs up to 8 are reported. However, Figures 7 and 11 (experimental data) include 11 and 12 IMFs respectively. Perhaps, a different simulated signal, resulting in a more similar number of modes, would serve better the paper, for comparability between numerical and experimental results.
2. the paper has been submitted to the S.I. "Optical Fiber Sensor Technology for Structural Health Monitoring" but there does not seem to be any reference to Optical Fiber in the text. Please motivate this choice.
3. CEEMDAN is used for signal decomposition. How is this specific design choice justified? E.g. in https://doi.org/10.3390/s21051825 it was proved that, at least for some specific signal processing applications, other alternatives such as variational mode decomposition (VMD) clearly outperforms CEEMDAN for damage detection. This aspect should be addressed in the paper.
4. For the artificially inserted measurement noise: how has the 20 dB case been selected? which is the rationale for it rather than a lower/higher amount of artificially added noise? Also, it would be more useful to test several decreasing levels of signal-to-noise ratios, to investigate the actual limits of the proposed methodology.
5. Overall, the figures are grainy and have a low resolution; please replace them with higher resolution / higher quality ones.
6. Please double-check your text since there are some typos and grammar errors. E.g. in the caption of Figure 3, there is an unnecessarily large blank space.
7. Figure 5: text and arrows superimposed to the text may help to better understand the experimental setup
8. Conclusions should also report some quantitative data, summarising the Results section
9. please fix the format of the Reference List according to MDPI guidelines for Authors
Author Response
Dear reviewer:
Thank you very much for your comments and suggestions on our articles. Please see the attachment for details of the modifications to the paper and the responses to your suggestions.
Yours Sincerely
Chunlan Mo, Huanyu Yang, Guannan Xiang, Guanjun Wang, Wei Wang, Xinghang Liu
and Zhi Zhou

Reviewer 2 Report
The article entitled "Monitoring bridge displacements based on BDS measurement with CEEMDAN-adaptive threshold wavelet method" is an interesting method that the author describes, but it is a more or less well-known method. The author refers to a reference, which unfortunately I was unable to open and check. Since I myself deal with this field and have consistently studied and tested the matter on many objects, it is difficult to capture uniform results with the GNSS method, especially the result for further analysis. Only with further analysis by FFT or Lomb transform can we arrive at a dynamic response. Even these results are subject to conditional use due to uneven and too slow sampling.
The leveling model is well derived and from the final results the values for calculating the dynamic response can be obtained.
However, I have a few concerns and comments:
1. In the introduction, I miss the comparison with previous research and what is the goal of this work?
2. Chapter 2 should be devoted to data collection methods and presented data collection problems
3. The chapter Principle of CEEMDAN-Adaptive Threshold Wavelet algorithm… is difficult for the reader to understand. I miss one short introduction to this chapter, where the author should present the algorithm a little better and in more detail
4. I didn't find out who the equations were based on
5. In this part, there is no explanation of what the individual labels in the equations mean
This chapter is written very unprofessionally and incomprehensible. I don't see the meaning of this subsection.
6. Unedited text in lines 120,121,122
7. Equation 7: Where does 0.6745 come from?
8. To verify the proposed denoising method, this paper used:…. Why is this method used?
9. Chapter 3 could be described in more detail as this is the point of the article!
10. The conclusion is modest, it should be better presented. It should be described what they achieved with this research, what can still be achieved and where are the limitations of the research?
11. Nowhere is it stated about the accuracy of this method or, to a large extent, this method is reliable for determining the dynamic response of the structure. Is 100Hz a high enough frequency to determine the response? We tested it ourselves and found that the GNSS method for determining the height of displacements is too modest and has too much noise in the results, which can greatly disturb further analyses.
12. It might be reasonable to use some other relevant method in a specific case, for example RTS, inductive meters, vibrometer or accelerated and then make a comparison with one of the listed methods. This is how it is usually done in the engineering and scientific profession.
13. Reference so too modest! A lot of research has been done in this area. The author should check works such as:
- Measurement of the dynamic displacements and of the modal frequencies of a short-span pedestrian bridge using GPS and an accelerometer F Moschas, S Stiros - Engineering structures, 2011 – Elsevier
- GPS/RTS data fusion to overcome signal deficiencies in certain bridge dynamic monitoring projects F Moschas, PA Psimoulis, SC Stiros
- Noise characteristics of high-frequency, short-duration GPS records from analysis of identical, collocated instruments F Moschas, S Stiros - Measurement, 2013 – Elsevier
- Monitoring dynamic and quasi-static deformations of large flexible engineering structures with GPS: Accuracy, limitations and promises A Nickitopoulou, K Protopsalti, S Stiros - Engineering Structures, 2006 –
- Measurement of bridge dynamic displacements and natural frequencies by RTS A Marendić, R Paar, D Damjanović - Građevinar, 2017
- Possibilities of Surveying Instruments in Determination of Buildings' Dynamic Displacements A Marendić, Z Kapović, R Paar - Geodetski list, 2013
- Processing of signals produced by strain gauges in testing measurements of the bridges B Kovačič, A Štrukelj, N Vatin - Procedia engineering, 2015 – Elsevier
- A Comparative Study of Signal Processing Methods for Contactless Geodetic Monitoring S Lubej, B Kovačič - Applied Sciences, 2021
- SYNCHRONISATION OF CONTACTLESS VIBRATION MONITORING METHODS B Kovacic, L Mursec, S Lubej - International Journal of Simulation …, 2022 - ijsimm.com
- Non-contact monitoring for assessing potential bridge damages
B Kovačič, L Muršec, S Toplak… - E3S Web of …, 2020 - e3s-conferences.org
- Dynamic deformation monitoring of a technological structure
A Kopáčik, I Lipták, P Kyrinovič, J Erdélyi - Geodetski list, 2013 – New trends of automate bridge monitoring
- A Kopacik, P Kyrinovic, J Erdélyi, I Lipták - Reports on Geodesy, 2011 - bibliotekanauki.pl
- Monitoring the response of Severn Suspension Bridge in the United Kingdom using multi‐GNSS measurements HA Msaewe, PA Psimoulis, CM Hancock… - … Health Monitoring, 2021 –
- Monitoring the dynamics of formby sand dunes using airborne LiDAR DTMs AMA Mahmoud, E Hussain, A Novellino, P Psimoulis… - Remote Sensing, 2021
Author Response

(The authors gave the same response as above.)

Round 2
Reviewer 1 Report
Overall, this Reviewer is satisfied with the replies made by the Authors to the most relevant issues, especially the content-related ones, and the changes they implemented in the manuscript.
The text is still hampered by some editorial issues, some typos, and grammar mistakes (e.g. page 9 line 315, there is a missing blank space between ‘wavelet’ and the following bracket; the same issue can be found on page 1 line 34 after ‘station’ and at line 38 after ‘errors’; Figure 12.a has some Chinese characters in the bottom left corner that could be better cropped out; etc.
However, these and other minor issues can also be corrected by the Authors at the proofreading stage.
Author Response
Dear reviewer:
Thank you for your valuable comments. We have carefully revised the manuscript according to your suggestion. Please check the attachment for the reply of specific modification content. Thanks again for your comments.
Best wishes.
Yours Sincerely
Chunlan Mo, Huanyu Yang, Guannan Xiang, Guanjun Wang, Wei Wang, Xinghang Liu
and Zhi Zhou
